


# Distinct responses of Asian summer monsoon to black carbon aerosols and greenhouse gases

Xiaoning Xie[1,2], Gunnar Myhre[3], Xiaodong Liu[1,4], Xinzhou Li[1], Zhengguo Shi[1], Hongli Wang[5], Alf Kirkevåg[6], Jean-Francois Lamarque[7], Drew Shindell[8], Toshihiko Takemura[9], and Yangang Liu[10]

[1]SKLLQG, Institute of Earth Environment, Chinese Academy of Sciences, Xi'an, China
[2]CAS Center for Excellence in Quaternary Science and Global Change, Xi'an, China
[3]Center for International Climate and Environmental Research, Oslo, Norway
[4]University of Chinese Academy of Sciences, Beijing, China
[5]School of Tourism and Hospitality Management, Shaanxi Radio and TV University, Xi'an, China
[6]Norwegian Meteorological Institute, Oslo, Norway
[7]National Center for Atmospheric Research, Boulder, USA
[8]Nicholas School of the Environment, Duke University, Durham, USA
[9]Climate Change Science Section, Kyushu University, Fukuoka, Japan
[10]Environmental and Climate Sciences Department, Brookhaven National Laboratory, Upton, NY, USA

*Correspondence to:* Xiaoning Xie (xnxie@ieecas.cn)

**Abstract.** Black carbon (BC) aerosols emitted from natural and anthropogenic sources induce positive radiative forcing and global warming, which in turn significantly affect the Asian summer monsoon (ASM). However, many aspects of the BC effect on ASM remain elusive and largely inconsistent among previous studies, which is strongly dependent on different low-level thermal feedbacks over the Asian continent and the surrounding ocean. This study examines the response of ASM to BC

5    forcing in comparison with the effect of doubled greenhouse gases (GHGs) by analyzing the Precipitation Driver Response Model Intercomparison Project (PDRMIP) simulations under an extreme high BC level (10 times modern global BC emissions or concentrations, labeled by BC×10) from nine global climate models (GCMs). The results show that although BC and GHGs both enhance the ASM precipitation minus evaporation (P−E) (a 13.6% increase for BC forcing and 12.1% for GHGs from the nine-model ensemble, respectively), there exists a much larger uncertainty in changes in ASM P−E induced by BC

10    than by GHGs. The summer P−E is increased by 7.7% to 15.3% due to these two forcings over three sub-regions including East Asian, South Asian, and western North Pacific monsoon regions. Further analysis of moisture budget reveals distinct mechanisms controlling the increases in ASM P−E induced by BC and GHGs. The change in ASM P−E by BC is dominated by the dynamic effect due to the enhanced large-scale monsoon circulation, whereas the GHG-induced change is dominated by the thermodynamic effect through increasing atmospheric water vapor. Radiative forcing of BC significantly increases the

15    upper-level atmospheric temperature over the Asian region to enhance the upper-level meridional land-sea thermal gradient (MLOTG), resulting in a northward shift of the upper-level subtropical westerly jet and an enhancement of the low-level monsoon circulation; whereas radiative forcing of GHGs significantly increases the tropical upper-level temperature, which reduces the upper-level MLOTG and suppresses the low-level monsoonal circulation. Hence, our results indicate a different mechanism of BC climate effects under the extreme high BC level, that BC forcing significantly enhances the upper-level





atmospheric temperature over the Asian region, determining ASM changes, instead of low-level thermal feedbacks as indicated by previous studies.

## 1   Introduction

Black carbon (BC) aerosols emitted from natural and anthropogenic sources absorb and scatter the shortwave radiation, heat

the air, and act as the second largest contributor to the current global warming after greenhouse gases (Hansen, et al., 2000; Jacobson, 2001; Forster, et al., 2007; Ramanathan and Carmichael, 2008; Mahajan et al., 2013; Bond, et al., 2013). In recent decades, with the explosive economic development and rapid population growth in India and China, large amounts of BC aerosols have been emitted to the atmosphere and have significantly affected the global and reginal energy balance, as well as climate, especially over South Asia and East Asia (Ramanathan and Carmichael, 2008; Boucher et al., 2013; Bond et al., 2013;

Li et al., 2016).

Many GCM investigations have indicated that BC aerosols can significantly influence the Asian summer monsoon (ASM) including South Asia summer monsoon (SASM) and East Asian summer monsoon (EASM). However, the effects of BC forcing on ASM remain elusive and largely inconsistent in previous studies, as they are strongly dependent on different low-level thermal feedbacks over the Asian continent and the surrounding ocean (e.g., Menon et al., 2002; Ramanathan and Carmichael,

2008; Bond et al., 2013; Boucher et al., 2013; Wu et al., 2015; Li et al., 2016). For SASM, Lau et al. (2006) showed that BC aerosols combined with natural dusts can increase precipitation over northern India and the Himalayan foothills through an elevated heat pump effect (EHP) during the pre-monsoon season. The EHP hypoethsis was corroborated by other modeling studies (Meehl et al., 2008). In summer, BC aerosols cause an increased cooling over the land surface relative to the adjacent ocean, decreasing the land-sea thermal gradient, weakening the SASM circulation and decreasing the associated precipitation

over the India subcontinent (Lau and Kim, 2007). On the other hand, other GCM studies do not support the monsoon weakening; instead they show that BC aerosols can enhance the atmospheric meridional heating gradient by increasing the low-level atmospheric temperature over South Asia, strengthening the monsoonal circulation with a stronger moisture flux into South Asia and an increase in the monsoonal rainfall (Wang, 2004; Ramanathan et al., 2005; Chung and Ramanathan, 2007). Studies with coupled GCMs with an interactive ocean showed that the BC-induced SST change over the India ocean also has significant

influences on SASM (Ramanathan et al., 2005; Chung and Ramanathan, 2007; Meehl et al., 2008). The decreased evaporation due to the cooling of North India Ocean owing to BC dimming can decrease the water moisture and the monsoonal inflow into South Asia, as well as the monsoonal precipitation (Ramanathan et al., 2005; Meehl et al., 2008). This BC-induced dimming over the North Indian Ocean can yield a decrease in meridional SST gradient which acts to weaken the monsoonal circulation and reduce the summer precipitation, as shown by numerous studies (Ramanathan et al., 2005; Chung and Ramanathan, 2007).

Additionally, there exist inconsistent results of the EASM changes induced by BC aerosols, as reviewed by Wu et al. (2015) and Li et al. (2016). Menon et al. (2002) claimed based on the GCM results with a high proportion of BC aerosols (the single scattering albedo (SSA) is given as a constant of 0.85) that anthropogenic aerosols can enhance the summer floods in southern China and increase the summer droughts over the northern China (southern-flood-northern-drought, labeled by SFND). How-





ever, the results cannot be reproduced with the higher and more realisitic SSA of 0.9 as derived from the satellite and surface observations (Lee et al., 2007). By considering both BC semidirect effect and indirect effect, BC aerosols have been shown to result in a reduction in clouds and precipitation in the southern China and an enhancement in rainfall in the northern China by enhancing the western Pacific subtropical high (Zhang et al., 2009), which is opposite to the SFND pattern. In general, the

effects of BC forcing on both SASM and EASM remain largely inconsistent, due to their strong dependency on differences in land or sea low-level thermal feedbacks to BC.

Furthermore, the comprehensive assessment including the vast majority of current GCMs shows that GCMs substantially underestimate the surface BC mass concentration, and its scattering and absorption optical thickness (by a factor of 2 to 4 over South Asia and East Asia), and thus severely underestimate the atmospheric heating and surface dimming of BC forcing,

mainly due to low biases of the regional emission inventory and inaccurate parameterizations of aerosol processes (Bond et al., 2013; Shindell et al., 2013; Pan et al., 2015; Fan et al., 2018). The underestimated BC concentration in GCMs may be an important factor responsible for inconsistent feedbacks of ASM mentioned above. Therefore, it is essential to investigate the feedbacks of ASM induced by BC using high BC concentration.

Therefore, the purpose of the present paper is to determine the effects on ASM of BC forcing under the extreme high

BC level, in comparison with the effect of a doubling of $CO_2$ concentration by analyzing the simulations of multiple GCMs in the Precipitation Driver Response Model Intercomparison Project (PDRMIP). PDRMIP provides the climate responses to individual forcers including extreme high BC level with 10 times modern global BC emissions or concentrations (Myhre et al., 2017). Previous studies (Samset et al., 2016; Stjern et al., 2017) have shown the importance of the globally averaged temperature and precipitation responses to BC forcing. Another objective is to use the PDRMIP data to see whether or not BC-

induced changes in ASM result from a distinct mechanism different from the above-mentioned one about low-level thermal dominated feedbacks of BC forcing.

The rest of this paper is structured as follows. Section 2 provides a brief description of the PDRMIP models, model setups and experiments, data, and the definition of the Asian monsoon domain. Section 3 shows results on the evaluation of the PDRMIP multi-model mean results and changes in ASM due to the BC and GHG forcings, respectively. In Section 4, the

physical mechanisms of changes in ASM induced by BC and GHGs are discussed including their effective radiative forcing (ERF) and atmospheric temperature changes at the surface and at higher altitudes. Further discussions are shown in Section 5, and the summary and conclusion are provided in Section 6.

## 2  PDRMIP data and Asian monsoon domain

### 2.1  Model description and PDRMIP data

In the PDRMIP project, global climate models coupled with ocean have run several numerical simulations with similar model configurations and forcing baseline, as well as perturbations with five climate drivers (Myhre et al., 2017). These GCMs have simulated a baseline experiment with modern aerosol emissions or concentrations and greenhouse gases with the year 2000 (denoted BASE). The numerical experiments of five perturbations were performed: a doubling of $CO_2$ concentrations (here-





after referred to CO2×2), 3 times CH4 concentrations (CH4×3), an increase in solar insolation with 2% (Solar +2%), 10 times modern global BC emissions or concentrations (BC×10), and 5 times modern global sulfate emissions or concentrations (SO4×5). The nine GCMs with BASE and the perturbation experiments are CanESM2, GISS-E2, HadGEM2, HadGEM3-GA4, IPSL-CM5A, MIROC-SPRINTARS, NCAR-CESM1-CAM4, NCAR-CESM1-CAM5, and NorESM1 for analyses, de-

tailed by Myhre et al. (2017). Additionally, several GCMs in PDRMIP from Liu et al. (2018) also simulated regional aerosol perturbation experiments e.g., 10 times modern BC or SO4 concentrations over the Asia region (labeled by BC×10ASIA and SO4×10ASIA, respectively).

Here, we analyze the PDRMIP data from BASE and the two climate perturbations including CO2×2 and BC×10 based on the nine GCMs. In terms of the BASE and other experiments with each perturbation, two kinds of numerical simulations

were performed: one using fixed SST (hereafter referred to as fSST) and the other one using a fully coupled ocean or a slab ocean (denoted coupled). The fSST experiments of at least 15 years and the coupled experiments of at least 100 years were run in these GCMs. In the following, the effective radiative forcing (hereafter ERF) is calculated as the net radiative fluxes changes (including shortwave+longwave radiative fluxes) at top of atmosphere (TOA) between perturbation experiments and BASE from the last 10 years of fSST experiments (Forster et al., 2016). The feedback response of climate including ASM

was diagnosed as the difference between the perturbation simulations and BASE based on the last 50 years of the coupled experiments. All modeled climate data including temperature, precipitation, and wind fields, as well as other physical variables were re-gridded into 2.5°×2.5° horizonal resolution for use in the present analysis in our study.

### 2.2 Definition of Asian monsoon domain

Based on the definition of the reference from Wang et al. (2012), the monsoon domain was defined as the region where the

differences between summer and winter rainfall are larger than 2 mm day$^{-1}$ and the local summer rainfall exceeds 55 percent of annual total amount. In the Northern Hemisphere, the local summer is defined as May-September (MJJAS) and the winter is November-March (NDJFM), and vice versa for the Southern Hemisphere. According to the above definition of monsoon domain, Figure 1 shows the spatial distribution of the Asian monsoon region based on the observed precipitation from 1979 to 2011 of Climate Prediction Center (CPC) merged analysis of precipitation (hereafter referred to CMAP) according to Xie

and Arkin (1997). The defined Asian monsoon region is further divided into three sub-regions including East Asian monsoon (105-160° E and 25-42.5° N), South Asian monsoon (60-105° E and 0-25° N), and western North Pacific monsoon (105-160° E and 0-25° N). We utilize the above defined Asian monsoon region and the sub-regions to calculate the regional average in the following analyses.



## 3 Results

### 3.1 Evaluation of multi-model results

In this subsection, we first examine the veracity of the simulated ASM by multiple-models using the sea level pressure (SLP), the low-level large-scale circulation, and the associated precipitation from the NCEP-DOE Reanalysis 2 (labeled by NCEP2)
(Kanamitsu et al., 2002) and the CMAP precipitation for 1979−2011. Figure 2 compares the summer (MJJAS) SLP, wind fields at 850 hPa, and the surface precipitation from observations and the multi-model mean (MMM) of the BASE experiment. The summer SLP field features a higher pressure over the low-latitude ocean and the western North Pacific ocean, and it has a relatively lower pressure over the Asian continent (Figures 2a and 2b), which leads to north-south and east-west land-sea thermal gradients. This spatial pattern of SLP induces a cross-equatorial flow at 850 hPa over low latitudes near the equator and
a south-westerly flow over the Indian subcontinent, the northern Bay of Bengal, and the Arabian sea, as well as southerly winds over the eastern China (Figure 2c and 2d). Figures 2e and 2f show larger precipitation over the coast of the Asian maritime continent and weaker precipitation over the Asian inland. Note that MMM of these simulations produces excessive rainfall over the southern slope of Tibetan Plateau compared to the CMAP precipitation, mainly due to a relatively coarse horizontal resolution of the models in Figure 2f. Therefore, MMM of the BASE experiment can mainly capture the most prominent
features of ASM in terms of the summer SLP, the 850 hPa large-scale circulation, and the surface precipitation in Figure 2.

Figure 3 shows the upper-level (200 hPa) geopotential height, wind fields, and westerly wind speeds from NCEP2 for 1979−2011 and MMM of the BASE experiment. Figure 3a shows the South Asian High near 30° N (or Tibetan High) in the upper-level at 200 hPa, producing a continental-scale anticyclonic outflow toward the low latitudes and the ocean, and driving the westerly upper-level jet near 40° N in NCEP2 (Figure 3c). This spatial pattern of the South Asian High and the upper-level
subtropical westerly jet is also depicted realistically in MMM of the BASE experiment, although the center of the South Asian High is stronger than that in NCEP2 (Figure 3b) and the intensity of the westerly jet is somewhat weaker (Figure 3d). In general, MMM of the BASE run can capture most monsoon features, including the low-level and high-level thermal structures and the large-scale circulation associated with monsoon, as well as the monsoonal precipitation as shown in Figures 2 and 3.

### 3.2 Comparison of ASM Changes induced by BC and GHGs

Figure 4 shows the domain-averaged value and spatial distribution of changes in MJJAS precipitation minus evaporation (hereafter denoted P−E, unit: mm day$^{-1}$) for BC and GHG forcings over the Asian monsoon region. Evidently, both BC and GHGs cause a substantial wetting over the Asian monsoon region with an increase in MMM of 0.37±0.28 mm day$^{-1}$ from BC forcing and 0.29±0.13 mm day$^{-1}$ from GHGs. There are a 13.6% increase for BC×10 and 12.1% for CO2×2 in this region (Table 1). Moreover, the increase in summer P−E due to BC forcing has a wider range from 0.01 (IPSL-CM5A) to
0.88 mm day$^{-1}$ (HadGEM2) for the individual GCMs, whereas it has a narrower range between 0.10 (IPSL-CM5A) to 0.47 mm day$^{-1}$ (NCAR-CESM1-CAM4) for GHGs. Therefore, the BC-induced changes in summer P−E have higher uncertainty in these models than that from GHGs according to the standard deviation of MMM and the range of P−E changes from the nine GCMs. The spatial distribution of BC and GHGs-induced changes in MMM P−E shows that the summer P−E are all





significantly increased over the three sub-regions including the East Asian, South Asian, and western North Pacific monsoon regions in Figures 4c and 4d. Domain-averaged calculated values in Table 1 indicate an increase in MMM P−E from 0.16 mm day$^{-1}$ (7.7%) to 0.48 mm day$^{-1}$ (15.3%) induced by these two forcings over these three sub-regions.

In order to investigate the mechanism governing the summer P−E responses for BC and GHG forcings, we have diagnosed
a moisture budget analysis assuming the linearized formulation as follows (Seager et al., 2010; Seager and Naik, 2012):

$$\Delta(P-E) = \Delta TH + \Delta DY + \Delta Res. \tag{1}$$

In the above equation, $\Delta(P-E)$ represents the term in terms of P−E changes between the sensitivity (perturbation) and control (BASE) experiments. The thermodynamic term $\Delta TH$ is influenced only by changes in specific humidity, with fixed large-scale circulations, whereas the dynamic term $\Delta DY$ involves changes in the large-scale circulation assuming no changes in specific
humidity. The residual term $\Delta Res$ includes transient eddy fluxes, surface quantities and nonlinear terms (Seager et al., 2010; Seager and Naik, 2012). Figure 5 shows the domain-averaged terms in MMM and the spatial distribution of each term in the moisture budget equation (1) during the summer season over the Asian monsoon domain. For BC forcing, the regional wetting is dominated by the dynamic term in Figure 5a, whereas the thermodynamic term contributes to the increase in ASM P−E slightly. From the spatial distribution of the thermodynamic and dynamic terms in Figures 5b and 5d, one can see a high and
statistically significant increase in the dynamic term but a low and insignificant increase in the thermodynamic term over the Asian monsoon region, which also indicates that the P−E change due to BC is dominated by the dynamic term. For GHG forcing, the increase in ASM P−E is controlled by the thermodynamic term whereas the dynamic term has a negative value (Figure 5a). Figures 5c and 5e also show a significant increase in the thermodynamic term over the Asian monsoon region, and an insignificant decrease in the dynamic term over most of the Asian monsoon region. Therefore, the moisture budget analysis
indicates that the ASM P−E increases due to BC and GHGs are dominated by distinct dynamic and thermodynamic terms, respectively.

Due to the absolute differences in the above moisture budget analysis between the BC and GHG forcings, we further analyze the BC and GHG-induced changes in the large-scale circulation (Figure 6) and specific humidity (Figure 7). It can be seen that BC forcing induces much more significant changes in the large-scale circulation compared to GHG forcing, including
low-level and high-level horizontal wind fields and vertical velocity. Figure 6a shows that BC forcing significantly increases the south-westerly flow over the Arabian sea and India, and enhances the southerly winds over the eastern China. It shows a significant increase in the upward vertical velocity at 500 hPa induced by BC over the Asian monsoon region (Figure 6c). These results indicate the significant enhancement of the large-scale ASM circulation induced by BC forcing. Additionally, changes in the large-scale circulation due to BC forcing are also evident in the spatial pattern of the upper-level westerly wind
in Figure 6e. It shows a significant decrease in westerly wind speed over the regions between 20°N and 40°N and an increase at higher latitudes, illustrating that BC forcing induces a northward movement of the upper-level subtropical westerly jet. This northward movement of the subtropical westerly jet is absolutely consistent with the low-level ASM circulation enhancement (Figure 6a). Figures 6b, 6d, and 6f show insignificant changes due to GHGs in low-level and high-level wind fields, and the vertical velocity at 500 hPa. Furthermore, Figure 7 shows the changes in MMM of MJJAS vertically integrated water





vapor due to the BC and GHG forcings. This figure shows that both BC and GHGs induce statistically significant increases in vertically integrated water vapor due to BC and GHG-induced atmospheric warming. However, the change in vertically integrated water vapor to GHGs is much larger than that to BC forcing. Therefore, BC yields a significant enhancement of large-scale ASM circulation and a relatively smaller increase in water vapor, implying that the dynamic term dominates the

ASM P−E increase and the thermodynamic term compensates its increase. However, GHGs induce a larger increase in water vapor and insignificant changes in the large-scale circulation, which suggests that the increase in ASM P−E due to GHGs is dominated by the thermodynamic term. These results indicate that the changes in the large-scale circulation and the water vapor due to BC and GHG forcings are consistent with the above moisture budget analysis.

## 4   Physical mechanisms

To seek physical mechanisms underlying the distinct responses of ASM to BC and GHGs, Figure 8 shows the spatial distribution of MMM values of the MJJAS ERF (W m$^{-2}$) induced by (a) BC and (b) GHGs over the Asia region, according to the corresponding fSST experiments in PDRMIP. The BC-induced ERF yields a significant positive value over the Asian continent, whereas it has an insignificant change over the surrounding oceans including the Indian ocean and Pacific ocean (Figure 8a). The much larger positive forcing particularly over the Indian subcontinent (larger than +5 W m$^{-2}$) and over the eastern China

(exceeding +10 W m$^{-2}$) are mainly due to existence of high BC emissions and loadings over these two regions. In contrast, GHG-induced ERF is significantly positive and uniform (below +7.5 m$^{-2}$) over the Asian continent and the surrounding oceans (Figure 8b), mainly because GHGs are well-mixed on a global scale. Our analysis suggests that there are obvious differences in the spatial distribution between BC and GHG-induced ERF, although both of them induce positive radiative forcings at the TOA.

Figures 9 shows the BC and GHG-induced changes in MMM of MJJAS surface atmospheric temperature and SLP. It can be seen that BC induces a greater surface warming over the Eurasian continent at middle and high latitudes and a smaller surface warming over low latitude ocean regions (Figure 9a). Note that the surface temperature in MMM has insignificant decreases over the Indian subcontinent and increases with a smaller amplitude over the eastern China. However, there also exists different sign (increase or decrease) in surface temperature changes due to BC over both these two regions under the extreme high BC

level for the individual models (Figure S1), consistent with the previous PDRMDIP study by Stjern et al. (2017). These results are also consistent with previous results based on present-day BC concentrations level (Bond et al., 2013; Wu et al., 2015; Li et al., 2016). Hence, there are inter-model differences in low-level temperature feedbacks due to BC over South Asia and East Asia under not only present-day but also extreme high BC concentration levels. Figure 9c shows that a significant decrease in SLP exists over the Eurasian continent at middle and high latitudes due to the BC-induced surface warming over this region.

However, the changes in SLP are small and insignificant over the Indian subcontinent and the eastern China because of smaller changes in surface temperature. Hence, the BC-induced increases in the surface thermal and pressure gradients between land and ocean are insignificant (Figures 9a and 9c). GHGs induce a surface global warming with a greater warming over the whole continent and a lesser warming over ocean due to different heat capacity of land and ocean (Fig. 9b). Figure 9d shows





a significant decrease of SLP over the Eurasian continent and a significant increase over the surrounding ocean, indicating an increased land-sea thermal and pressure gradients due to GHG forcing at the surface.

In order to compare with the low-level thermal feedbacks, Figures 10 shows the BC and GHG-induced upper-level changes in (a, b) MMM of MJJAS atmospheric temperature and (c, d) geopotential height at 200 hPa. Figure 10a shows that BC induces

the significant increase in the atmospheric temperature at 200 hPa over land and ocean. However, the upper-level atmospheric temperature increases are much larger over the Asian continent (especially over mid-latitude regions) compared to the tropics. The change in upper-level atmospheric temperature leads to a significant increase in 200 hPa geopotential height, especially at middle latitudes over the Asian continent (Figure 10c). The significant increases in upper-level temperature and geopotential height over the Asian continent result in the increase of the upper-level meridional land-ocean thermal gradient (MLOTG),

which makes the northward movement of the upper-level subtropical westerly jet (Figure 6e) and enhances the low-level ASM circulation (Figures 6a and 6c). GHGs indicate a more significant increase in 200 hPa atmospheric temperature over the tropics in Figure 10b, resulting in a much larger increase in 200 hPa geopotential height in Figure 10d. This spatial pattern of changes in atmospheric temperature and geopotential height at 200 hPa induced by GHGs decreases the upper-level MLOTG, which makes the low-level ASM circulation and the upper-level westerlies insignificantly change (Figures 6b, 6d, and 6f). Pronounced

warming in the tropical upper troposphere has been found from observational evidence (Fu et al., 2004; Santer et al., 2005) and GCMs (IPCC, 2007) with global warming, leading to decreases in the upper-level MLOTG and the low-level ASM circulation (Ueda et al., 2006; Endo et al., 2014). These changes in the temperature of the tropical upper troposphere and ASM circulation due to GHGs are the same as with our results.

Additionally, we also show the vertical profile of BC and GHG-induced changes in MMM of MJJAS MLOTG in Figure 11.

MLOTG is defined as the difference between the Asian region (60-125° E and 10-42.5° N) and low latitudes of ocean (60-125° E and −10-10° N). The MLOTG values have insignificant changes due to BC forcing at the low-levels, and notable increases with height and has a maximum at 200 to 300 hPa. The significant upper-level MLOTG increase due to BC forcing determines changes in the upper-level and low-level large-scale circulation over the Asian monsoon region. Although the GHG-induced MLOTG increase is evident at the low levels, whereas it decreases with height and has a minimum with negative values at 200

hPa. The decrease in upper-level MLOTG induced by GHGs forcing suppresses the large-scale ASM circulation. Therefore, these changes in the vertical profile of MLOTG due to BC and GHGs are absolutely consistent with the above results at the surface and upper level as shown in Figures 9 and 10.

## 5 Further discussions

The PDRMIP project also conducted the global sulfate perturbation experiments (hereafter referred to SO4×5) and the Asian

aerosol perturbation experiments including BC×10ASIA and SO4×10ASIA (Myhre et al., 2017; Liu et al., 2018). Figure 12 shows the domain-averaged value and spatial distribution of changes in the summer P−E over the Asian monsoon region due to (a, d) increasing Asian BC, (b, e) global SO4, and (c, f) Asian SO4. Figures 12a and 12d show that the Asian BC significantly increases ASM P−E in MMM of 0.18±0.13 mm day$^{-1}$ with a large range from 0.06 (IPSL-CM5A) to 0.37 mm



day$^{-1}$ (NCAR-CESM1-CAM5). The moisture budget analysis shows that the dynamic term dominates the increase in ASM P−E for increasing Asian BC, whereas the thermodynamic term almost has nothing to contribute to the increase in Figures S3a, S3b, and S3c. The Asian BC significantly enhances the low-level monsoon circulation enhancement (Figures S3d and S3e) and makes a northward shift of upper-level subtropical westerly jet (Figure S3f). Hence, the trends in the ASM P−E

and low-level and upper-level circulations due to the increasing Asian BC are very similar to the response from global BC in Figures 4 and 6. Note that the increase in ASM P−E due to the Asian BC is much lower than that from the global BC (0.37±0.28 mm day$^{-1}$). That is mainly because that the thermodynamic term has no insignificant increases induced by Asian BC due to smaller increases in atmospheric water vapor (Figure S3g), compared to that of global BC (Figure 7a). Additionally, the increasing global (Figures 12b and 12e) and Asian SO4 (Figures 12c and 12f) both significantly reduce the ASM P−E in

MMM of −0.41±0.18 and −0.56±0.22 mm day$^{-1}$, which is of opposite sign to BC and GHGs (Figure 4). The increasing SO4 yields a significant negative ERF over the Asian region (Figures S4b and S4c), which is reverse to that of BC (Figures 8a and S4a). The negative ERF due to SO4 remarkably decreases the surface temperature over the Asian region, which results in a reduction in the land-sea thermal gradient and thus reduces the Asian summer monsoonal circulations and the associated precipitation (Liu et al., 2011, Xie et al., 2016). The decrease in MMM of the ASM P−E due to the increasing Asian SO4 is

larger than that of the global SO4, that is mainly because that PDRMIP runs the Asian sulfate perturbation experiment with 10 times modern SO4 over Asia and the global sulfate perturbation experiment only with 5 times modern global SO4.

Relative to GHGs shown in Figure 4b, the modeled BC-induced change in MJJAS P−E has a much larger uncertainty over the Asian monsoon region for the individual GCMs (Figures 4a and 12a), more likely due to intermodel differences in the regional ERF. Figure 13 shows the global and Asian BC-induced MJJAS domain-averaged ERF over the Asian region for

individual models and MMM, and regression of the domain-averaged change in ASM P−E versus ERF, also including that of the global and Asian SO4. Figure 13a shows that MMM of ERF over the Asian region is 5.00±1.71 W m$^{-2}$ and 4.75±2.31 W m$^{-2}$ for the global and Asian BC, and −8.41±2.95 W m$^{-2}$ and −12.81±3.81 W m$^{-2}$ for the global and Asian SO4, respectively. Further analysis in Figure 13b illustrates that there exists a significant positive correlation between ASM P−E change and the regional ERF for global ($R^2$=0.83) and regional aerosols including BC and SO4 ($R^2$=0.90), respectively. This

positive correlation indicates that the aerosol-induced ERF over the Asian region mainly dominates the ASM P−E changes for the individual GCMs, where larger positive (negative) ERF increases (decreases) the ASM P−E more substantially. The corresponding main reason is that larger positive ERF due to BC over the Asian region more significantly heats the regional atmosphere and increases land-ocean thermal gradient, leading to the ASM circulation enhancement and the monsoonal P−E increase, and larger negative ERF induced by SO4 has reverse results. Note that the slope of the regression line for global

aerosols is larger (0.058 mm day$^{-1}$ *per* W m$^{-2}$) than that of regional aerosols (0.042 mm day$^{-1}$ *per* W m$^{-2}$), mainly because that the thermodynamic term compensates the the ASM P−E changes for the increasing global aerosols. Additionally, we also check the relationship between the ASM P−E and the regional ERF over Asia (or the global ERF) for GHGs, indicating that there are no positive correlations between them (not shown). In general, accurate descriptions of physical and radiative processes associated with BC climate forcing will reduce the uncertainty of BC-induced ERF and ASM precipitation changes

(Bond et al., 2013).





Menon et al. (2002) reported that BC forcing can induce the SFND pattern over the eastern China from GCMs, while other GCM simulations do not support this spatial pattern of summer precipitation changes (e.g., Lee et al., 2007; Zhang et al., 2009; Wang et al., 2017). In Figure 14, we also show the spatial distribution of MMM MJJAS precipitation changes due to the global (BC×10) and regional BC (BC×10ASIA), which indicates a similar spatial pattern for these two forcings. The

results in MMM show a tripole pattern of precipitation changes over the eastern China: increase over the northern China, the southern China, and the adjacent ocean, and a decrease over the middle and lower reaches of the Yangtze River. This tripole pattern of precipitation changes over the eastern China is mainly caused by the enhancement of the low-level ASM circulation and the northward movement of the upper-level westerly jet. Similar patterns have been shown to take place in modern and paleoclimate based on observations and models (Chiang et al., 2015; 2017; Kong et al., 2017).

Additionally, Figure S1 shows the changes induced by BC forcing in the MJJAS surface atmospheric temperature at 2m for the individual models. BC forcing produces an increase or a decrease in the surface temperature based on the nine GCMs over both the Indian subcontinent and the eastern China. The GCMs of MIROC-SPRINTARS, NCAR-CESM1-CAM4, NCAR-CESM1-CAM5, and NorESM1 indicate an significant decrease in surface temperature over the India subcontinent, whereas other GCMs show an insignificant reduction or increase in the surface temperature. The GISS-E2, HadGEM2 and MIROC-

SPRINTARS show a reduction in the surface temperature over the eastern China, and other GCMs show an increase in surface temperature including CanESM2, HadGEM3, IPSL-CM5A, NCAR-CESM1-CAM4, NCAR-CESM1-CAM5, and NorESM1. Hence, the sign of changes in surface temperature due to BC forcing is strongly dependent on different GCMs over South Asian and East Asian, which is absolutely consistent with previous investigations (Menon et al., 2002; Ramanathan and Carmichael, 2008; Boucher et al., 2013; Bond et al., 2013). It indicates the complexity of low-level temperature feedbacks due to BC over

these two regions, which can lead to distinct responses of ASM (Bond et al., 2013; Wu et al., 2015; Li et al., 2016). However, under this extreme high BC level, it shows more significant and larger increase in the high-level atmospheric temperature over the Asian continent compared to the surrounding ocean, which is true for all the nine models (Figure S2). These upper-level thermal feedbacks due to BC forcing result in a northward shift of upper-level subtropical westerly jet and an enhancement of low-level monsoon circulations in the individual GCMs (not shown), which is very similar to the MMM results as shown in

Figure 6. Hence, under even more extreme high BC levels, the ASM changes induced by BC forcing would be more consistent between the individual GCMs due to pronounced high-level heating of BC over the Asian continent.

## 6 Concluding remarks

This study examines the response of ASM due to BC forcing under the extreme high BC level (BC×10) from the nine GCMs based on the PDRMIP project, compared to GHGs with CO2×2. Our analyses show that BC and GHGs enhance the ASM

P−E by 0.37±0.28 mm day$^{-1}$ (13.6%) and 0.29±0.13 mm day$^{-1}$ (12.1%) from the nine-model ensemble, respectively. These results show that there exists a larger uncertainty in changes in ASM P−E induced by BC than by GHGs. The summer P−E are all increased with a range between 7.7% and 15.3% induced by these two forcings over the three sub-regions including East Asian, South Asian, and western North Pacific monsoon regions. It shows that distinct mechanisms determine the changes in





ASM P−E induced by BC and GHGs. The change in P−E by BC is dominated by the dynamic effect due to the enhanced large-scale monsoon circulation, whereas the thermodynamic effect mainly determines the change by GHGs through the increased atmospheric water vapor. BC forcing significantly increases the upper-level temperature over the Asian region to enhance the upper-level MLOTG, leading to a northward shift of upper-level subtropical westerly jet and an enhancement of low-

level monsoon circulations, whereas radiative forcing of GHGs significantly increases the upper-level temperature over the low-latitude ocean, which reduces the upper-level MLOTG and suppresses the ASM circulation. Hence, our results reveal a new mechanism of BC climate effects: under the extreme high BC level, BC forcing significantly heats the upper-level atmosphere and thus increases the upper-level temperature over the Asian region, which determines changes in the upper-level westerlies and the ASM circulation and associated precipitation. Additionally, further analysis shows that there exists a

significant positive correlation between ASM P−E change and aerosol-induced ERF from the individual models, indicating that the aerosol-induced ERF mainly dominates the changes in ASM P−E. Therefore, accurate descriptions of the associated physical and radiative processes of BC climate forcing will reduce the uncertainty of BC-induced ERF and ASM changes (Bond et al., 2013).

Two points are noteworthy. First, the impact of snow darkening by deposition of BC and dust aerosols on snow cover over the

Tibetan Plateau (TP) indicates a significant positive radiative forcing and regional surface warming (Flanner et al., 2007; Qian et al., 2015), which then will enhance TP thermal effects and affect the ASM circulation and precipitation (Qian et al., 2011; 2015; Xie et al., 2018a, 2008b; Shi et al., 2019). In the PDRMIP project, there are two types of GCMs including the emission-driven models (CanESM2, HadGEM2, MIROC-SPRINTARS, and NCAR-CESM1-CAM5) and the concentration-driven models (GISS-E2, HadGEM3-GA4, IPSL-CM5A, NCAR-CESM1-CAM4, and NorESM1) in the experiments of BC×10 (Samset

et al., 2016; Myhre et al., 2007; Stjern et al., 2017). However, we cannot isolate the effects of BC-induced snow darkening over TP on ASM from the total effects under this extreme BC level based on both the concentration-driven and emission-driven models. Hence, additional experiments considering BC direct and snow-darkening effects separately are needed to solve this problem in the future. Second, BC can serve as ice nuclei (IN) and thus significantly affects the global and regional climate via the interactions of BC with snow and ice, although it accounts for relatively small percentage of aerosol mass in the at-

mosphere. However, there exists a larger uncertainty in the aerosol indirect effect associated with aerosol-cloud interactions (Bond et al., 2013). It is essential to accurately parameterize these aerosol-cloud processes in GCMs for further assessment of the BC climate effects, although some GCMs (MIROC-SPRINTARS, NCAR-CESM-CAM5, and NorESM1) in PDRMIP includes effects of BC on snow and ice (Stjern et al., 2017).

*Data availability.* All the data about the PDRMIP project used for our manuscript can be derived from the Norwegian

NORSTORE data storage facility for the public by http://www.cicero.uio.no/en/PDRMIP/PDRMIP-data-access.

*Competing interests.* The authors declare that they have no conflict of interest.

*Acknowledgements.* Xiaoning Xie was partially funded by the National Natural Science Foundation of China (Grant No. 41991254), the Strategic Priority Research Program of Chinese Academy of Sciences (Grant No. XDB40030100), the National Key Research and Development Program of China (Grant No. 2016YFA0601904), and the CAS "Light of West China" Program (Grant No. XAB2019A02). Alf





Kirkevåg was supported by the Research Council of Norway (grant nos. 229771, 285003, and 285013), and by Notur/NorStore (NN2345K and NS2345K). Yangang Liu was supported by the U.S. Department of Energy's Atmospheric System Research (ASR) program.





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



**Table 1.** Multi-model mean of the MJJAS precipitation minus evaporation (P−E, unit: mm day$^{-1}$) in the Asian monsoon region (AM), and the sub-regions including East Asian monsoon (EAM), South Asian monsoon (SAM), and western North Pacific monsoon (WPM) for BASE, BC×10, and CO2×2 experiments, as well as corresponding differences between BC×10 (or CO2×2) and the BASE experiment.

| Regions | BASE | BC×10 | CO2×2 | Difference (BC×10−Base) | Difference (CO2×2−Base) |
|---------|------|-------|-------|-------------------------|-------------------------|
| AM region | 2.73 | 3.10 | 3.06 | 0.37 (13.6%) | 0.33 (12.1%) |
| EAM region | 2.07 | 2.23 | 2.30 | 0.16 (7.7%) | 0.23 (11.1%) |
| SAM region | 3.13 | 3.61 | 3.43 | 0.48 (15.3%) | 0.31 (9.9%) |
| WPM region | 2.60 | 2.95 | 2.88 | 0.35 (13.5%) | 0.28 (10.8%) |



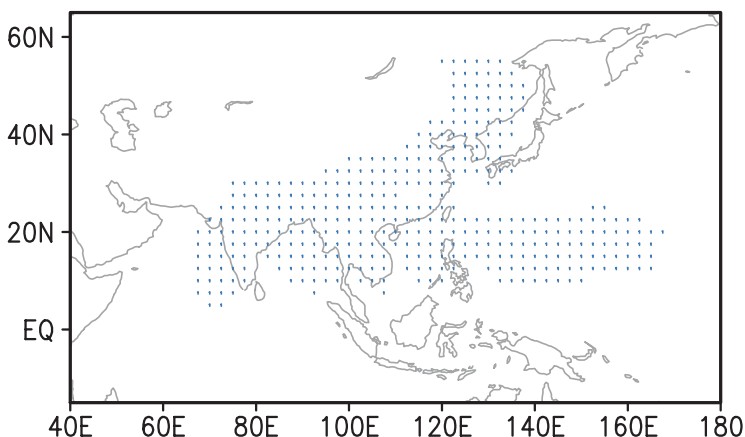

**Figure 1.** Spatial distribution of the Asian monsoon region (stippled, blue) based on the CMAP data from 1979-2011.

**Figure 2.** Comparison of the climatological MJJAS sea level pressure (SLP, unit: hPa), the wind field at 850hPa (UV850, unit: m s$^{-1}$), and surface precipitation (P, unit: mm day$^{-1}$) between (a, c, e) observations from 1979-2011 (NCEP2 wind field, and CMAP precipitation) and (b, d, f) muti-model mean of the BASE experiments.

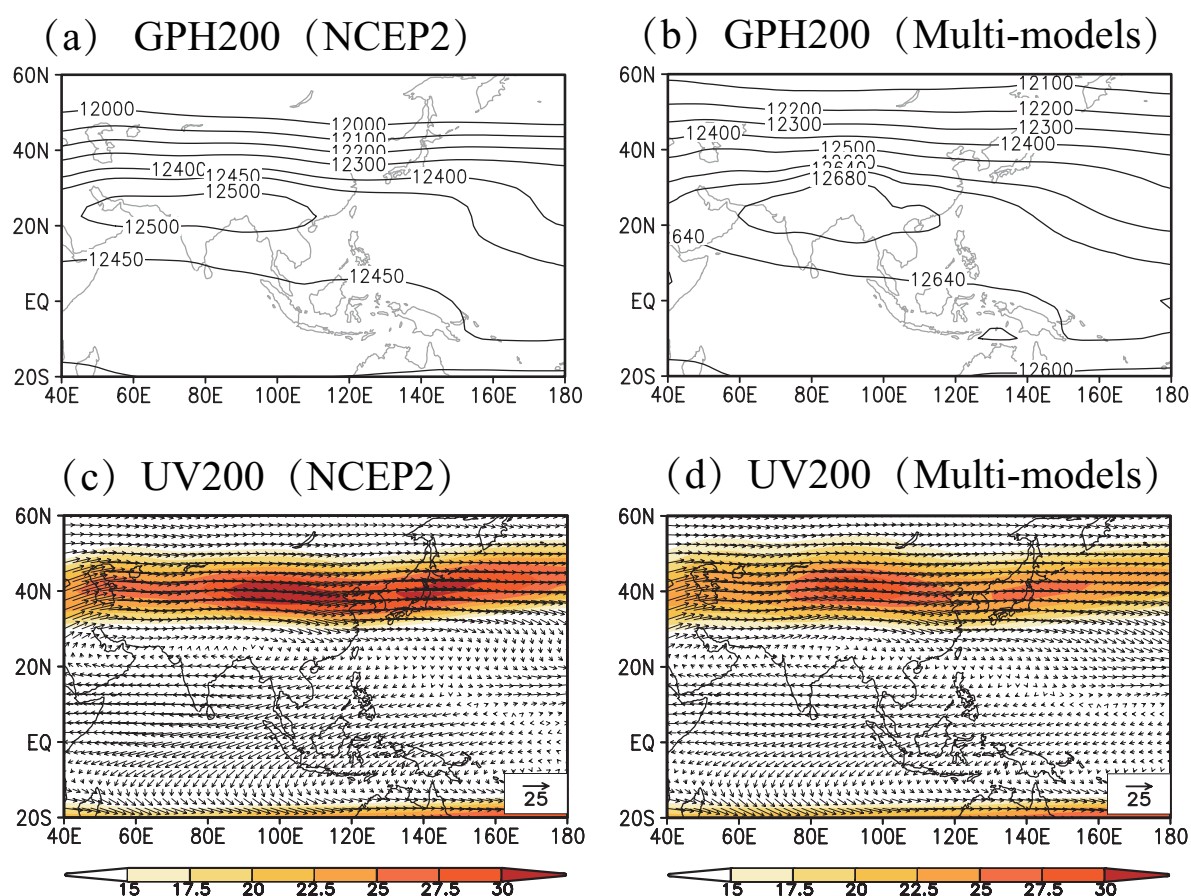

**Figure 3.** Comparison of the climatological MJJAS mean of 200 hPa geopotential height (GPH200, unit: gpm), and 200 hPa wind field (UV200, unit: m s$^{-1}$) and westerly wind speed wind speed (shaded, unit: m s$^{-1}$) between (a, c) observations from 1979-2011 and (b, d) muti-model mean of the BASE experiments.

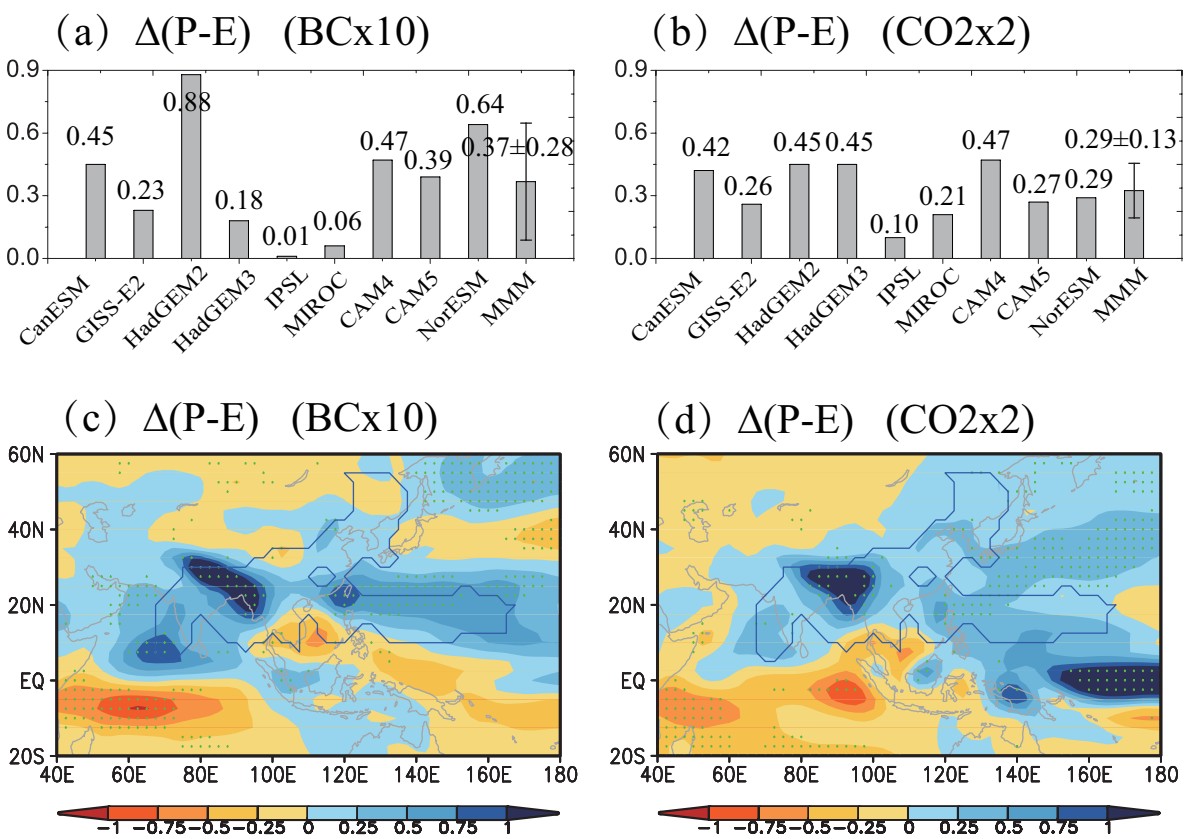

**Figure 4.** Changes in the MJJAS precipitation minus evaporation ($\Delta$(P−E), unit: mm day$^{-1}$) under (a, c) increasing BC and (b, d) GHG. (a, b) Domain-averaged value over the Asian monsoon region, and (c, d) spatial pattern of the multi-model mean (MMM). Error bars (a, b) of MMM represent the standard deviation. Dotted regions (c, d) indicate where MMM is more than 1 standard deviation away from zero. The areas within the blue line represent the Asian monsoon region.



**Figure 5.** (a), MJJAS domain-averaged changes (unit: mm day$^{-1}$) in the multi-model mean (MMM) precipitation minus evaporation ($\Delta$(P−E)), the thermodynamic term ($\Delta$TH), the dynamic term ($\Delta$DY), and residual term ($\Delta$Res) of moisture budget equation under increasing BC and GHG. (b, c) Spatial distribution of MMM MJJAS $\Delta$TH, (d, e) $\Delta$DY under increasing BC and GHG. Error bars (a) of MMM represent the standard deviation. Dotted regions (b, c, d, e) indicate where MMM is more than 1 standard deviation away from zero, and the areas within the blue line represent the Asian monsoon region.







**Figure 6.** Changes in (a, b) multi-model mean (MMM) of the MJJAS 850 hPa wind field ($\Delta$UV850, unit: m s$^{-1}$), (c, d) 500 hPa vertical velocity ($\Delta$Omega, unit: $0.01 \times$Pa s$^{-1}$)), and (e, f) 200 hpa westerly wind speed ($\Delta$U200, unit: m s$^{-1}$) under increasing BC and GHG. Black arrows (a, b) and dotted regions (c, d, e, and f) indicate where MMM is more than 1 standard deviation away from zero.



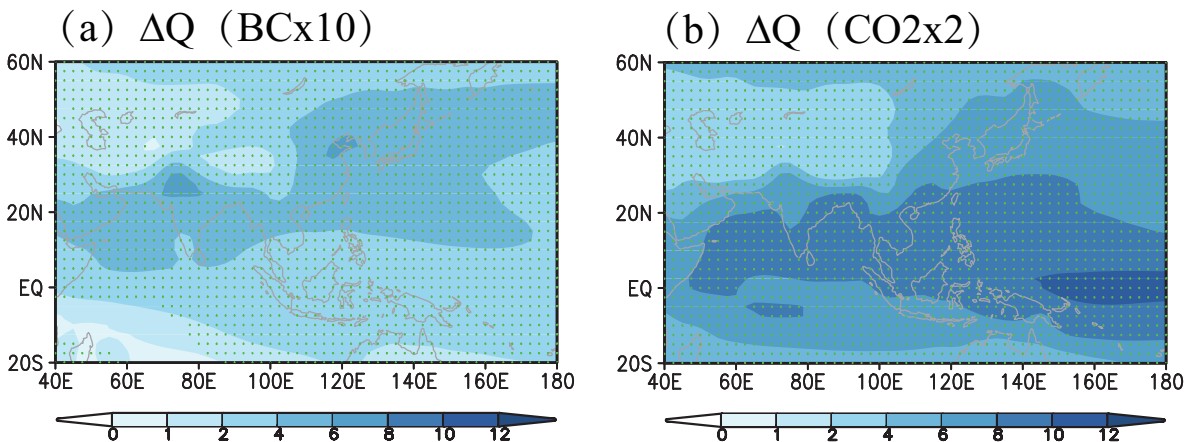

**Figure 7.** Changes in multi-model mean (MMM) of the MJJAS vertically integrated water vapor ($\Delta Q$, unit: g m$^{-2}$) under (a) increasing BC and (b) GHG. Dotted regions indicate where MMM is more than 1 standard deviation away from zero.



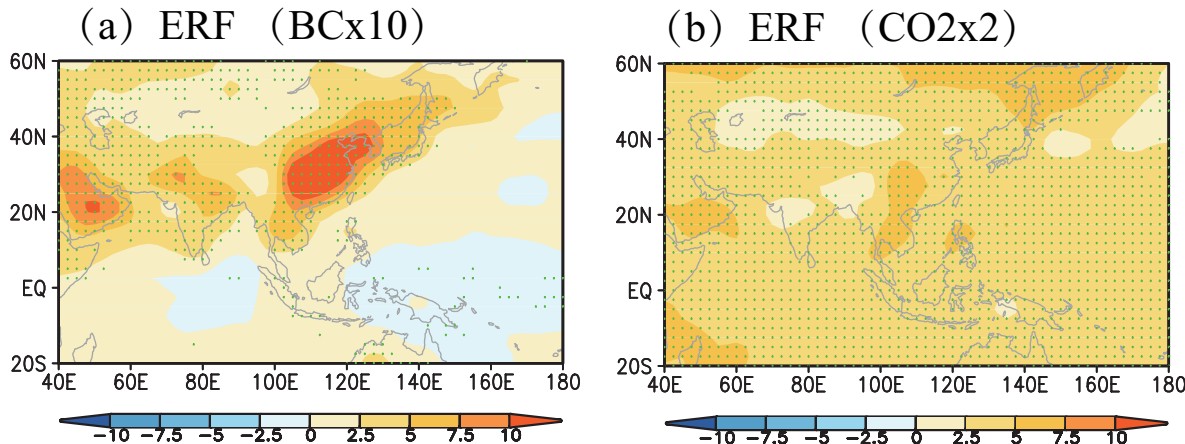

**Figure 8.** Multi-model mean (MMM) of the MJJAS effective radiative forcing (ERF, unit: W m$^{-2}$) under (a) increasing BC and (b) GHG. Dotted regions indicate where MMM is more than 1 standard deviation away from zero.

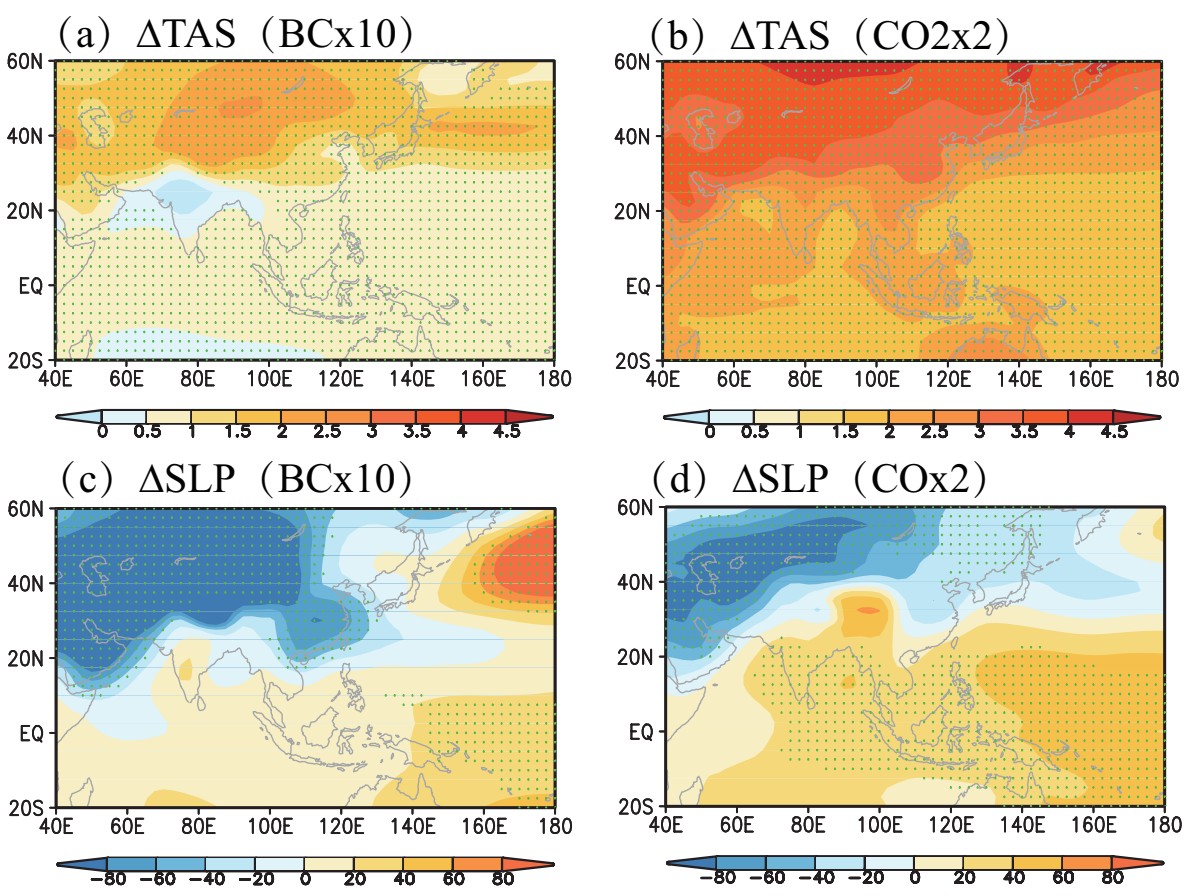

**Figure 9.** Changes in (a, b) multi-model mean (MMM) of the MJJAS surface atmospheric temperature (ΔTAS, unit: °C) and (c, d) sea level pressure (ΔSLP, unit: hPa) under increasing BC and GHG. Dotted regions indicate where MMM is more than 1 standard deviation away from zero.



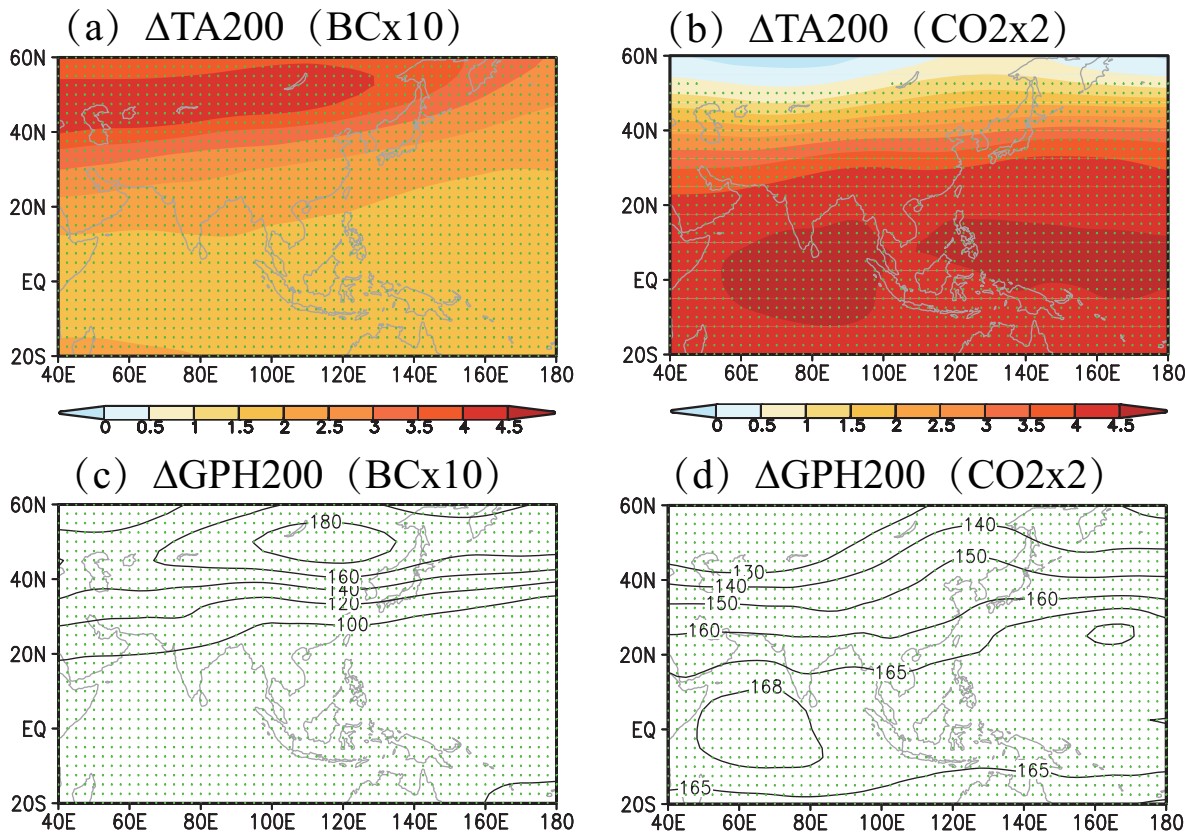

**Figure 10.** Changes in (a, b) multi-model mean (MMM) of the MJJAS 200 hPa atmospheric temperature (ΔTA200, unit: °C) and (c, d) 200 hPa geopotential height (ΔGPH200, unit: gpm) under increasing BC and GHG. Dotted regions indicate where MMM is more than 1 standard deviation away from zero.

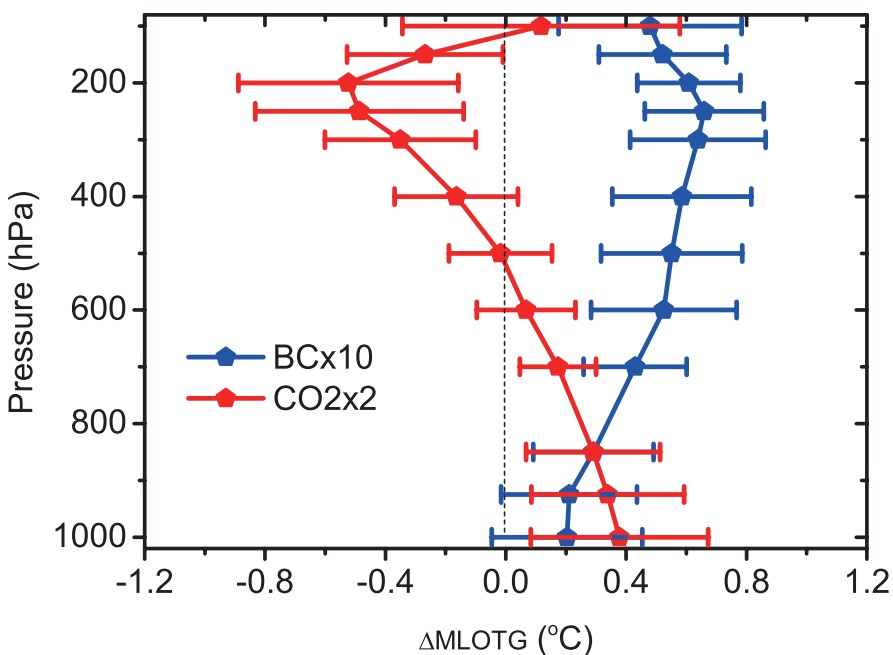

**Figure 11.** Changes in multi-model mean (MMM) of the MJJAS meridional land-ocean thermal gradient (ΔMLOTG, unit: °C) under increasing BC and GHG. The MLOTG is defined as the difference between Asian region (60−125° E and 10−42.5° N) and ocean (60−125° E and −10−10° N). Error bars of MMM indicate the standard deviation.



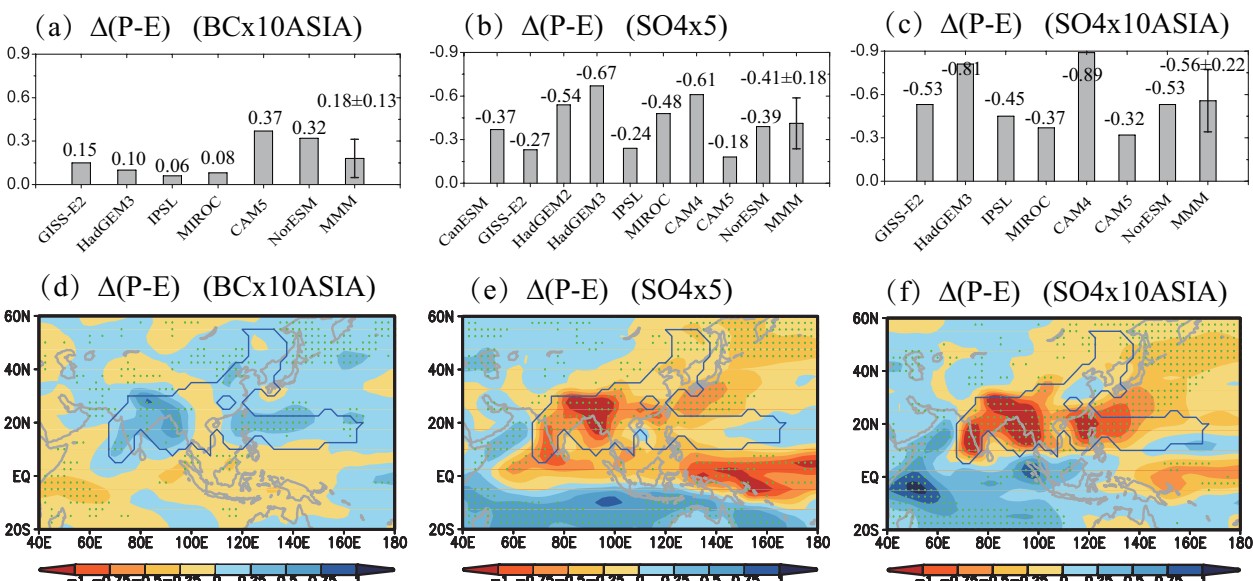

**Figure 12.** Changes in the MJJAS precipitation minus evaporation ($\Delta(P-E)$, unit: mm day$^{-1}$) under (a, d) increasing Asian BC, (b, e) global SO4, and (c, f) Asian SO4. (a, b, c) Domain-averaged value over the Asian monsoon region, and (d, e, and f) spatial pattern of the multi-model mean (MMM). Error bars (a, b, and c) of MMM represent the standard deviation. Dotted regions (d, e, and f) indicate where MMM is more than 1 standard deviation away from zero. The areas within the blue line represent the Asian monsoon region.

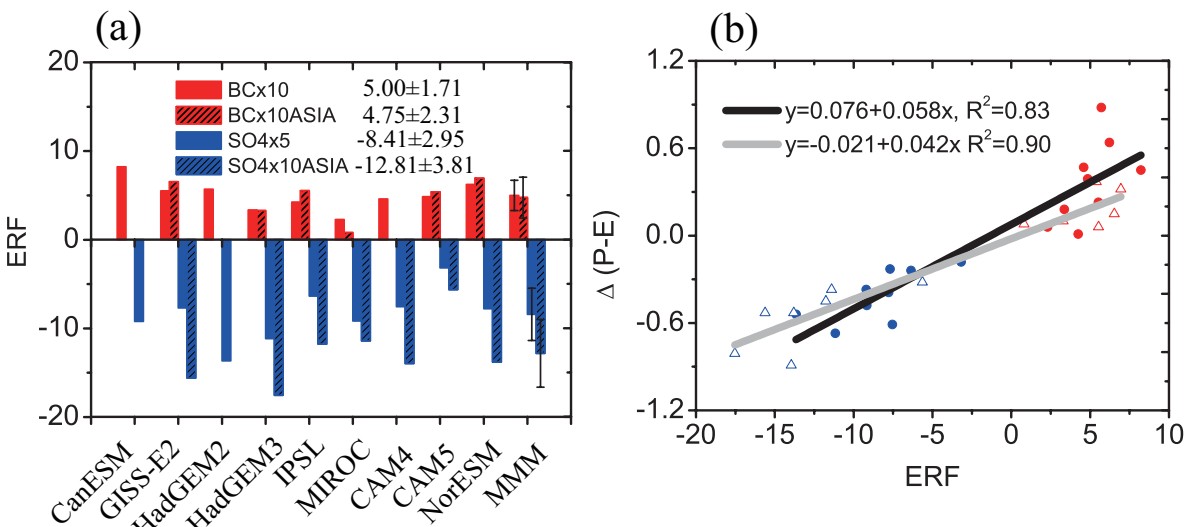

**Figure 13.** (a), MJJAS domain-averaged effective radiative forcing over the Asian region with 60-125° E and 10-42.5° N (ERF, unit: W m$^{-2}$) under increasing global (BC×10 and SO4×5) and Asian aerosols (BC×10ASIA and SO4×10ASIA), where error bars of multi-model mean (MMM) represent the standard deviation. (b), Regression of the domain-averaged change in MJJAS precipitation minus evaporation over the Asian monsoon region (Δ(P−E), unit: mm day$^{-1}$) versus the regional ERF for global aerosols (black line) and for Asian aerosols (gray line). Red and blue circles represent the global BC and SO4, and red and blue triangles indicate the Asian BC and SO4, respectively.

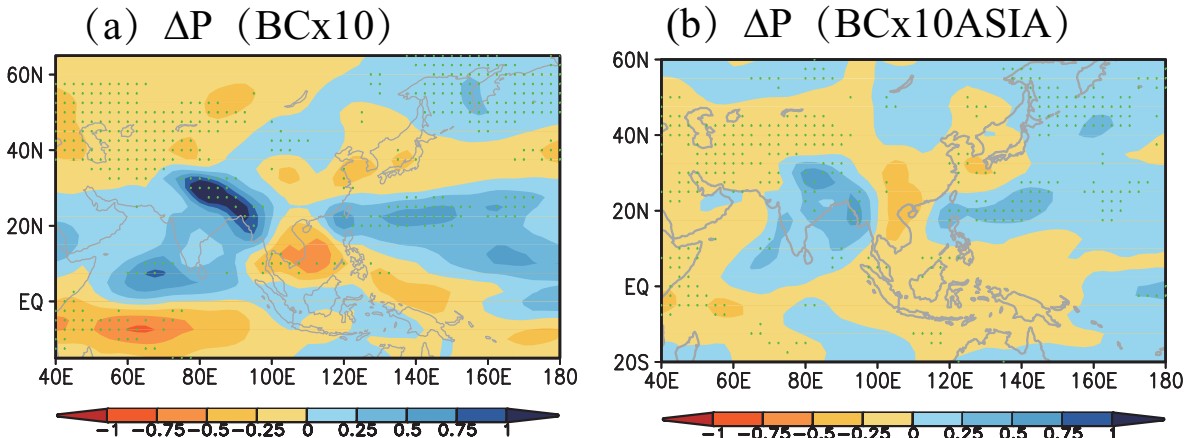

**Figure 14.** Changes in multi-model mean (MMM) of the MJJAS precipitation ($\Delta P$, unit: mm day$^{-1}$) under (a) increasing global (BC$\times$10) and (b) Asian BC (BC$\times$10ASIA). Dotted regions indicate where MMM is more than 1 standard deviation away from zero.