# Peer review of "Distinct responses of Asian summer monsoon to black carbon aerosols and greenhouse gases"

_Atmospheric Chemistry and Physics, 2020_

## Referee Comment (RC1) · Anonymous Referee #1 · 27 Jun 2020

Review's comments for the paper: ACP-2020-483, entitled "Distinct responses of Asian summer monsoon to black carbon aerosols and greenhouse gases", submitted to ACP

General comments

By using PDRMIP simulations under double greenhouse gases (GHGs) forcing and 10 times of present-day Black carbon (BC) forcing this paper investigates the changes of Asian summer monsoon (ASM). Results show that both GHGs changes and BC changes lead to enhanced precipitation minus evaporation over the Asian monsoon regions, but physical processes involved show some distinct characteristics. GHG changes lead to enhanced monsoon precipitation mainly through the thermodynamic effect through increased water vapor in the atmosphere while changes by BC are through dynamical effect by enhanced large scale monsoon circulation due to en-

hanced upper tropospheric warming over Asia. The topic is an interesting one. Results are interesting and they are well described. The paper is worth of publication. However, there are some specific comments listed below that need to be addressed to improve the quality of the paper. The paper, therefore, needs a minor revision before it can be acceptable for publication.

Specific comments

1. Section 2.1 on pages 3-4. It is better to summarize experiments in a Table and to give some extra information about model horizontal and vertical resolutions. 2. Section 2.2 on page 4. Highlight the three monsoon regions in figure 1. 3. Line 26 on page 4Please confirm that the EASM region is over region (105-160E). It includes large part of ocean. 4. Line 6 on page 5. Change "the surface precipitation" to "precipitation" 5. Line 9 on page 5. Change "thermal gradients" to "pressure gradients" since this statement is on changes in SLP. 6. Lines 12-13. The statement of excessive precipitation over the southern slope of Tibetan Plateau in model simulations is due to model horizontal resolution lacks evidence to support this. Please add some refs to support it or whether you have analyzed individual model simulations to get this conclusion. 7. Line 15 on page 5. Change "the surface precipitation" to "precipitation" 8. Line 6 on page 6. The moisture budget decomposition is vertically integrated quantities. It is better to have word "vertically integrated" when authors describe each terms. 9. Line 20 on page 7. BC absorbs solar radiation and therefore leads to decreased solar radiation at the surface. Why it leads to warming of surface air temperature? Some clarifications on this would improve the paper and helpful for readers. 10. Line 33 on page 7. There are a rich of literatures on enhanced land warming over continents in response to GHG changes and these studies suggest that warming contrast is not due to different heat capacity of land and ocean. Suggest authors rephrase this and add a few of refs. 11. Line 14 on page 8. Rephrase this statement. 12. Line 3 on page 9. Remove word "enhancement".

---

## Referee Comment (RC2) · Anonymous Referee #2 · 19 Jul 2020

This study compares the influence of BC and GHGs on Asian summer monsoon based on the PDRMIP simulations, and the physical mechanisms that influence the responses are discussed as well. The topic is really interesting and the manuscript is well organized and presented, while I think the manuscript can be further improved by considering . I suggest the paper to be published after a major revision, and my specific comments are listed below:

1. Three sub-regions are defined for discussions, and dots with different colors are suggested to differ the three regions.

2. It is interesting and expected to find the uncertainties related to the BC×10 is larger than those of CO2×2, and would the authors give more discussions on the possible of the uncertainties?

[Figure]

3. CMAP precipitation and the corresponding references should be given. Is there also precipitation in NCEP2, and why different "observations" are used in Figure 2 (even if the variables are different)?

4. I found Section 4 really interesting, and would really suggest the authors to extend the corresponding discussions. For example, the authors simply mentioned that "Our analysis suggests that there are obvious differences in the spatial distribution between BC and GHG-induced ERF, although both of them induce positive radiative forcings at the TOA" for Figure 8. How the differences are introduced, and how such differences would further influence the ASM? Maybe the spatial distributions of BC and CO2 concentration differences introduce the differences. Thus, I suggest to include the BC and CO2 concentration distribution in the figure as well. This is not directly related to this study, but may be helpful to better understand the forcing. Meanwhile, I noticed that there are some regions with negative forcing, and how such forcing is introduced?

This is just one example, and I suggestion the section to be discussed in more details. However, this is just my personal suggestion, and it is totally up to the authors' choices.

5. The spatial variations of the variables should be better discussed. Maybe the standard deviation over space can be discussed and given as well.

6. SO4×5 is used, whereas, for the ASIA case, the SO4×10ASIA is considered, which makes the comparison less meaningful.

7. Labels for the markers should be given in Figure 13(b)

---

## Author Comment (AC1) · 27 Aug 2020

Manuscript Number: **ACP-2020-483**

Journal: ACP

The revised manuscript entitled "Distinct responses of Asian summer monsoon to black carbon aerosols and greenhouse gases" by Xiaoning Xie, Gunnar Myhre, Xiaodong Liu, Xinzhou Li, Zhengguo Shi, Hongli Wang, Alf Kirkevåg, Jean-Francois Lamarque, Drew Shindell, Toshihiko Takemura, and Yangang Liu.

We thank the ACP Handing Editor (Professor Jianzhong Ma) for his hard work and the two anonymous referees for their constructive suggestions to improve our manuscript significantly. We greatly appreciate the generally positive comments from both the two Reviewers (Reviewer #1 and Reviewer #2), and have addressed all the concerns, with point-by-point responses detailed below (reviewers comments in red color and our responses in blue color). If you have any questions, please do not hesitate to contact me via email at xnxie@ieecas.cn. Thank you very much for your kindness and hard work.

Best wishes,

Xiaoning Xie, Gunnar Myhre, Xiaodong Liu, Xinzhou Li, Zhengguo Shi, Hongli Wang, Alf Kirkevåg, Jean-Francois Lamarque, Drew Shindell, Toshihiko Takemura, and Yangang Liu.

Response to Reviewer #1:

General comments:

By using PDRMIP simulations under double greenhouse gases (GHGs) forcing and 10 times of present-day Black carbon (BC) forcing this paper investigates the changes of Asian summer monsoon (ASM). Results show that both GHGs changes and BC changes lead to enhanced precipitation minus evaporation over the Asian monsoon regions, but physical processes involved show some distinct characteristics. GHG changes lead to enhanced monsoon precipitation mainly through the thermodynamic effect through increased water vapor in the atmosphere while changes by BC are

through dynamical effect by enhanced large scale monsoon circulation due to enhanced upper tropospheric warming over Asia. The topic is an interesting one. Results are interesting and they are well described. The paper is worth of publication. However, there are some specific comments listed below that need to be addressed to improve the quality of the paper. The paper, therefore, needs a minor revision before it can be acceptable for publication.

Response: Thank the Reviewer #1 very much for the positive comments. According to the comments, we have addressed all the specific comments with point-by-point responses detailed below.

Specific comments

1. Section 2.1 on pages 3-4. It is better to summarize experiments in a Table and to give some extra information about model horizontal and vertical resolutions.

Thank the Reviewer for his suggestions. According to the Reviewer's comments, additional table (Table S1) has been added in the Supporting Information about the model horizontal and vertical resolutions, as well as aerosol information for the nine PDRMIP GCMs.

Table S1: Models used for the present study as summarized in Myhre e t al., (2017).

| Model | Version | Horizontal resolutions | Vertical resolutions | Ocean setup | Aerosol emissons |
|---|---|---|---|---|---|
| CanESM2 | 2010 | 2.8x2.8 | 35 levels | Coupled | Emissions |
| GISS-E2R | E2-R | 2x2.5 | 40 levels | Coupled | Fixed concentration |
| HadGEM2 | 6.6.3 | 1.875x1.25 | 38 levels | Coupled | Emissions |
| HadGEM3 | GA 4.0 | 1.875x1.25 | 85 levels | Coupled | Fixed concentration |
| IPSL-CM5A | 5A | 3.75 x1.875 | 19 levels | Coupled | Fixed concentration |
| MIROC-SPRINTARS | 5.9.0 | T85 | 40 levels | Coupled | Emissions |
| NCAR-CESM1-CAM4 | 1.0.3 | 2.5x1.9 | 26 levels | Slab ocean | Fixed concentration |
| NCAR-CESM1-CAM5 | 1.1.2 | 2.5x1.9 | 30 levels | Coupled | Emissions |
| NorESM1 | 1-M | 2.5x1.9 | 26 levels | Coupled | Emissions |

2. Section 2.2 on page 4. Highlight the three monsoon regions in figure 1.

Taken. According to the Reviewer's comment, the three monsoon regions are shown including East Asian, South Asian, and western North Pacific monsoon regions in Figure 1.

[Figure]

Figure 1. Spatial distribution of the Asian monsoon region (stippled, blue) including East Asian, South Asian, and western North Pacific monsoon regions based on the CMAP data from 1979-2011.

3. Line 26 on page 4 Please confirm that the EASM region is over region (105-160E). It includes large part of ocean.

Yes, the EASM region is defined as the region (105-160E) with dotted regions in Figure 1. However, there is not large dotted region over large part of ocean. Hence, the EASM does not include large part of ocean.

4. Line 6 on page 5. Change "the surface precipitation" to "precipitation".
Taken.

5. Line 9 on page 5. Change "thermal gradients" to "pressure gradients" since this statement is on changes in SLP.
Taken.

6. Lines 12-13. The statement of excessive precipitation over the southern slope of Tibetan Plateau in model simulations is due to model horizontal resolution lacks evidence to support this. Please add some refs to support it or whether you have analyzed individual model simulations to get this conclusion.

Yes, we have added the corresponding reference in the revised manuscript. Some researches show that low-resolution GCMs are inadequate to reproduce the precipitation closely associated with fine-scale orographic forcing, such as the narrow large-rainfall belt along the southern edge of the Tibetan Plateau. Spatial distribution of precipitation over and around the elevation of the Tibetan Plateau and high-altitude mountains becomes more realistic at higher resolutions (Li et al., 2015). Hence, we have added in the manuscript "Note that MMM of these simulations produces excessive rainfall over the southern slope of Tibetan Plateau compared to the CMAP precipitation, mainly due to a relatively coarse horizontal resolution of the models in Figure 2f, as noted by Li et al., (2015). They also show that spatial distribution of precipitation over and around the Tibetan Plateau and high-altitude mountains becomes more realistic for the higher resolutions (Li et al., 2015)."

7. Line 15 on page 5. Change "the surface precipitation" to "precipitation".

Taken.

8. Line 6 on page 6. The moisture budget decomposition is vertically integrated quantities. It is better to have word "vertically integrated" when authors describe each terms.

Yes, the Reviewer's point is correct. It indicates that the moisture budget decomposition is vertically integrated quantities. We have added the "vertically integrated" in the manuscript.

9. Line 20 on page 7. BC absorbs solar radiation and therefore leads to decreased solar radiation at the surface. Why it leads to warming of surface air temperature? Some clarifications on this would improve the paper and helpful for readers.

As noted by the Reviewer's comment, the BC aerosols can absorb solar radiation and therefore leads to decreased solar radiation at the surface. The rapid temperature response is a significant cooling as shown in fixed SST experiments over India, China, and Central Africa in Stjern et al., (2017), caused by the strong reduction in shortwave radiation reaching the surface. However, the increases in surface temperature in Figure 9 and Figure S1 are derived from the coupled experiments. Previous studies show that the change in surface temperature in the coupled experiments is dependent on the effective radiative forcing (ERF) at the TOA (Gregory et al., 2004; Myhre, 2013). Hence, the increase in surface temperature in Figure 9 and Figure S1 is resulted from positive ERF at the TOA in Figure 8a. The corresponding descriptions have been added in the manuscript.

10. Line 33 on page 7. There are a rich of literatures on enhanced land warming over continents in response to GHG changes and these studies suggest that warming contrast is not due to different heat capacity of land and ocean. Suggest authors rephrase this and add a few of refs.

Taken. We agree with the Reviewer's point and have added the corresponding

references (Lambert and Chiang 2007; Lambert et al., 2011) in the revised manuscript.

11. Line 14 on page 8. Rephrase this statement.

Taken. This sentence has been revised as "This spatial pattern of changes in atmospheric temperature and geopotential height at 200 hPa induced by GHGs decreases the upper-level MLOTG, leading to insignificant changes in the low-level ASM circulation and the upper-level westerlies (Figures 6b, 6d, and 6f)."

12. Line 3 on page 9. Remove word "enhancement".

Taken.

References

Lambert, F. H., and Chiang, J. C. H., Control of land-ocean temperature contrast by ocean heat uptake, Geophys. Res. Lett., 34, L13704, https://doi.org/10.1029/2007GL029755, 2007.

Lambert, F. H., Webb, M. J., and Joshi, M., The relationship between land-ocean surface temperature contrast and radiative forcing, J. Climate, 24(13), 3239-3256, https://doi.org/10.1175/2011JCLI3893.1, 2011.

Li, J., R. Yu, W. Yuan, H. Chen, W. Sun, and Y. Zhang, Precipitation over East Asia simulated by NCAR CAM5 at different horizontal resolutions, J. Adv. Model. Earth Syst., 7, 774–790, doi:10.1002/2014MS000414, 2015.

Gregory, J. M., W. J. Ingram, M. A. Palmer, G. S. Jones, P. A. Stott, R. B. Thorpe, J. A. Lowe, T. C. Johns, and K. D. Williams, A new method for diagnosing radiative forcing and climate sensitivity, Geophys. Res. Lett., 31, L03205, https://doi.org/:10.1029/2003GL018747, 2004.

Myhre, G., Shindell, D., Bréon, F.-M., Collins, W., Fuglestvedt, J., Huang, J., Koch, D., Lamarque, J.-F., Lee, D., Mendoza, B., Nakajima, T., Robock, A., Stephens, G., Takemura, T., and Zhang, H.: Anthropogenic and natural radiative forcing, in: Climate change 2013: The Physical Science Basis, Contribution of Working Group I

to the Fifth Assessment Report of the Intergovernmental Panel on Climate Change, edited by: Stoker, T. F., Qin, D., Plattner, G.-K., Tignor, M., Allen, S. K., Boschung, J., Nauels, A., Xia, Y., Bex, V., and Midgley, P. M., Cambridge University Press, Cambridge, UK and New York, USA, 2013.

Myhre, G., Forster, P., Samset, B., Hodnebrog, Ø, Sillmann, J., Aalbergsjø, S. G., Andrews, T., Boucher, O., Faluvegi, G., and Flächner, D.: PDRMIP: A precipitation driver and response model intercomparison project, protocol and preliminary results, B. Am. Meteorol. Soc., 98, 1185–1198, https://doi.org/10.1175/BAMS-D-16-0019.1, 2017.

Stjern, C. W., Samset, B. H., Myhre, G., Forster, P. M., Hodnebrog,, Andrews, T., Boucher, O., Faluvegi, G., Iversen, T., Kasoar, M., Kharin, V., Kirkevag, A., Lamarque, J.-F., Olivie, D., Richardson, T., Shawki, D., Shindell, D., Smith, C., Takemura, T., and Voulgarakis, A.: Rapid adjustments cause weak surface temperature response to increased black carbon concentrations, J. Geophys. Res.-Atmos., 122, 11462--1481, https://doi.org/10.1002/2017JD027326, 2017.

---

## Author Comment (AC2) · 27 Aug 2020

Manuscript Number: **ACP-2020-483**

Journal: ACP

The revised manuscript entitled "Distinct responses of Asian summer monsoon to black carbon aerosols and greenhouse gases" by Xiaoning Xie, Gunnar Myhre, Xiaodong Liu, Xinzhou Li, Zhengguo Shi, Hongli Wang, Alf Kirkevåg, Jean-Francois Lamarque, Drew Shindell, Toshihiko Takemura, and Yangang Liu.

We thank the ACP Handing Editor (Professor Jianzhong Ma) for his hard work and the two anonymous referees for their constructive suggestions to improve our manuscript significantly. We greatly appreciate the generally positive comments from both the two Reviewers (Reviewer #1 and Reviewer #2), and have addressed all the concerns, with point-by-point responses detailed below (reviewers comments in red color and our responses in blue color). If you have any questions, please do not hesitate to contact me via email at xnxie@ieecas.cn. Thank you very much for your kindness and hard work.

Best wishes,

Xiaoning Xie, Gunnar Myhre, Xiaodong Liu, Xinzhou Li, Zhengguo Shi, Hongli Wang, Alf Kirkevåg, Jean-Francois Lamarque, Drew Shindell, Toshihiko Takemura, and Yangang Liu.

Response to Reviewer #2:

General comments:

This study compares the influence of BC and GHGs on Asian summer monsoon based on the PDRMIP simulations, and the physical mechanisms that influence the responses are discussed as well. The topic is really interesting and the manuscript is well organized and presented, while I think the manuscript can be further improved by considering. I suggest the paper to be published after a major revision, and my specific comments are listed below.

Response: Thank Reviewer #2 very much for the positive comments and constructive

suggestions. We have addressed all the specific comments with point-by-point responses listed below.

1. Three sub-regions are defined for discussions, and dots with different colors are suggested to differ the three regions.

Taken. According to the Reviewer's comment, the three monsoon regions are shown including East Asian, South Asian, and western North Pacific monsoon regions in Figure 1.

[Figure]

Figure 1. Spatial distribution of the Asian monsoon region (stippled, blue) including East Asian, South Asian, and western North Pacific monsoon regions based on the CMAP data from 1979-2011.

2. It is interesting and expected to find the uncertainties related to the BC_10 is larger than those of CO2_2, and would the authors give more discussions on the possible of the uncertainties?

Yes, we have added some discussions on the possible reasons of more uncertainties due to BCx10 experiments. This larger uncertainty of BCx10 is mainly due to large uncertainty in BC-induced ERF as shown in Figure 13. This positive correlation in Figure 13b indicates that the aerosol-induced ERF over the Asian region mainly

dominates the ASM P-E changes for the individual GCMs, where larger positive (negative) ERF increases (decreases) the ASM P-E more substantially. Hence, the larger uncertainty of aerosols in ASM P-E is mainly resulted from large uncertainty in ERF. We have added the corresponding descriptions in the revised manuscript.

3. CMAP precipitation and the corresponding references should be given. Is there also precipitation in NCEP2, and why different "observations" are used in Figure 2 (even if the variables are different)?

According to the Reviewer's comments, we have added the reference about the CPC Merged Analysis of Precipitation (CMAP) for 1979-2011 (Xie and Arkin, 1997). The CMAP precipitation is often used to validate the model precipitation. NCEP-DOE Reanalysis 2 (labeled by NCEP2) is an improved version of the NCEP Reanalysis I model that fixed errors and updated paramterizations of physical processes (https://psl.noaa.gov/data/gridded/data.ncep.reanalysis2.html). This reanalysis mainly includes 3-d wind field, mainly based on model to perform data assimilation, whereas it does not include the variable of precipitation. Hence, observations of precipitation are based on CMAP, whereas observations of wind field are from NCEP2 data.

4. I found Section 4 really interesting, and would really suggest the authors to extend the corresponding discussions. For example, the authors simply mentioned that "Our analysis suggests that there are obvious differences in the spatial distribution between BC and GHG-induced ERF, although both of them induce positive radiative forcings at the TOA" for Figure 8. How the differences are introduced, and how such differences would further influence the ASM? Maybe the spatial distributions of BC and CO2 concentration differences introduce the differences. Thus, I suggest to include the BC and CO2 concentration distribution in the figure as well. This is not directly related to this study, but may be helpful to better understand the forcing. Meanwhile, I noticed that there are some regions with negative forcing, and how such forcing is introduced? This is just one example, and I suggestion the section to be discussed in more details. However, this is just my personal suggestion, and it is

totally up to the authors' choices.

Thank the Reviewer for his constructive suggestions. The greenhouse gas $CO_2$ is well-mixed. Hence, the $CO_2$ concentration is almost the same for everywhere, leading to uniform radiative forcing. Spatial distribution of BC concentration in PDRMIP has been shown in Figure S1 from the Reference (Stjern et al., 2017), which is absolutely same as our results (because we used the same PDRMIP data). These exists larger BC burden over India, China, and Central Africa in Figure S1. As the Reviewer mentioned, this pattern leads to the similar spatial distribution of ERF in Figure 8. Additionally, instantaneous radiative forcing (IRF) and effective radiative forcing (ERF) are often used to describe aerosol radiative forcing in the IPCC AR5 terminology, where ERF is recommended in IPCC AR5 (Boucher et al., 2013). ERF is defined as the change in net radiative long-wave (LW) plus short-wave (SW) fluxes at TOA in the fSST simulations, which includes fast responses (e.g., cloud feedback, water vapor feedback, and so on). Therefore, IRF due to BC shows positive values for everywhere in Figure 5 from the Reference (Stjern et al., 2017). However, ERF indicate complex changes with negative forcing over several regions due to fast responses (Stjern et al., 2017).

[Figure]

Figure S1. BCx10 minus BASE BC burden changes for the median of all nine models (left), the five models using concentration-based perturbation simulations (middle) and the four models using emission-based perturbation simulations (right). This Figure is absolutely from Figure S1 based on PDRMIP (Stjern et al., 2017).

5. The spatial variations of the variables should be better discussed. Maybe the standard deviation over space can be discussed and given as well.

Spatial distribution of MJJAS △(P-E) and the corresponding standard deviation due to BCx10AISA, SO4x5, and SO4x10ASIA was shown in Figure S2. The figure shows similar spatial pattern for △(P-E) and its standard deviation, where larger values in △(P-E) corresponds to larger standard deviation. Hence, its standard deviation can provide additional information. The increase in P-E in BCx10ASIA is true over almost the East Asian monsoon region, whereas the decreases in P-E are shown for SO4x5 and SO4x10ASIA over this region. However, spatial distribution of △(P-E) is inconsistent mainly due to regional dynamic responses related to complexity of Asian summer monsoon, as shown in many references e.g., Zhou et al. (2009).

[Figure]

Figure S2. Changes in the MJJAS precipitation minus evaporation (△P-E), unit: mm day$^{-1}$ and the corresponding standard deviation (STDEV) for (a, d) increasing Asian BC, (b, e) global SO4, and (c, f) Asian SO4. Dotted regions (a, b, and c) indicate where MMM is more than 1 standard deviation away from zero. The areas within the blue line represent the Asian monsoon region.

6. SO4_5 is used, whereas, for the ASIA case, the SO4_10ASIA is considered, which makes the comparison less meaningful.

Firstly, the experiments of SO4x5 and SO4x10ASIA are performed in the PDRMIP project in Table S2 (Myhre et al., 2017; Liu et al., 2018). We only show the results

about aerosol experiments about SO4x5 and SO4x10ASIA in PDRMIP project (Figure 12). Additionally, it also shows the regression of the P-E versus the regional ERF for global aerosols and for Asian aerosols in Figure 13b. This regression makes the comparisons between SO4x5 and SO4x10ASIA meaningful.

Table S2 Model simulations about aerosols analyzed in the current study.

| Experiment | BCx10 | BCx10ASIA | SO4x5 | SO4x10ASIA |
|---|---|---|---|---|
| Specifications | BC increased by 10 times globally | BC over Asia increased by 10 times | SO4 increased by 5 times globally | BC over Asia increased by 10 times |

7. Labels for the markers should be given in Figure 13(b)

Thank the Reviewer for his suggestions. Labels for the markers have been added in the Figure 13b.

[Figure]

Figure 13. (a), MJJAS domain-averaged effective radiative forcing over the Asian region with 60-125E and 10-42.5N (ERF, unit: W m$^{-2}$) under increasing global (BCx10 and SO4x5) and Asian aerosols (BCx10ASIA and SO4x10ASIA), where error bars of multi-model mean (MMM) represent the standard deviation. (b), Regression of the domain-averaged change in MJJAS precipitation minus evaporation over the Asian monsoon region ($\triangle$(P-E), unit: mm day$^{-1}$) versus the regional ERF for global aerosols (Reg.1) and for Asian aerosols (Reg.2).

References

Boucher, O., Randall, D., Artaxo, P., Bretherton, C., Feingold, G., Forster, P., Kerminen, V.-M., Kondo, Y., Liao, H., Lohmann, U., Rasch, P., Satheesh, S. K., Sherwood, S., Stevens, B., and Zhang, X. Y.: Clouds and aerosols, in: Climate change 2013: the physical science basis, Contribution of Working Group I to the Fifth Assessment Report of the Intergovernmental Panel on Climate Change, edited by: Stoker, T. F., Qin, D., Plattner, G.-K., Tignor, M., Allen, S. K., Boschung, J., Nauels, A., Xia, Y., Bex, V., and Midgley, P. M., Cambridge University Press, Cambridge, UK and New York, USA, 2013.

Liu, L., Shawki, D., Voulgarakis, A., Kasoar, M., Samset, B. H., Myhre, G., Forster, P. M., Hodnebrog, Ø, Sillmann, J., Aalbergsjø, S. G., Boucher, O., Faluvegi, G., Iversen, T., Kirkevåg, A., Lamarque, J., Olivié, D., Richardson, T., Shindell, D., and Takemura, T.: A PDRMIP Multimodel Study on the Impacts of Regional Aerosol Forcings on Global and Regional Precipitation, J. Climate, 31, 4429–4447, https://doi.org/10.1175/JCLI-D-17-0439.1, 2018.

Myhre, G., Forster, P., Samset, B., Hodnebrog, Ø, Sillmann, J., Aalbergsjø, S. G., Andrews, T., Boucher, O., Faluvegi, G., and Flächner, D.: PDRMIP: A precipitation driver and response model intercomparison project, protocol and preliminary results, B. Am. Meteorol. Soc., 98, 1185–1198, https://doi.org/10.1175/BAMS-D-16-0019.1, 2017.

Stjern, C. W., Samset, B. H., Myhre, G., Forster, P. M., Hodnebrog, Ø, Andrews, T., Boucher, O., Faluvegi, G., Iversen, T., Kasoar, M., Kharin, V., Kirkevåg, A., Lamarque, J.-F., Olivieì, D., Richardson, T., Shawki, D., Shindell, D., Smith, C., Takemura, T., and Voulgarakis, A.: Rapid adjustments cause weak surface temperature response to increased black carbon concentrations, J. Geophys. Res.-Atmos., 122, 11462–1481, https://doi.org/10.1002/2017JD027326, 2017.

Xie, P., and Arkin, P. A., Global precipitation: A 17-year monthly analysis based on gauge observations, satellite estimates, and numerical model outputs, B. Am. Meteorol. Soc., 78, 2539-2558,

https://doi.org/10.1175/1520-0477(1997)078<2539:GPAYMA>2.0.CO;2, 1997.

Zhou, T. J., Gao, D. Y., Li, J., and Li, B.: Detecting and understanding the multi-decadal variability of the East Asian Summer Monsoon—Recent progress and state of affairs, Meteorol. Z., 18, 455–467, 2009.